

# Secondary organic aerosol phase behaviour in chamber photo-oxidation of mixed precursors

Yu Wang[1], Aristeidis Voliotis[1], Yunqi Shao[1], Taomou Zong[2], Xiangxinyue Meng[2], Mao Du[1], Dawei Hu[1], Ying Chen[3,#], Zhijun Wu[2,4,5], M. Rami Alfarra[1,6], Gordon McFiggans[1,*]

5  [1]Centre for Atmospheric Science, Department of Earth and Environmental Sciences, The University of Manchester, Manchester M13 9PL, UK

[2]State Key Joint Laboratory of Environmental Simulation and Pollution Control, International Joint Laboratory for Regional Pollution Control, Ministry of Education (IJRC), College of Environmental Sciences and Engineering, Peking University, Beijing 100871, China

10  [3]Lancaster Environment Centre, Lancaster University, LA1 4YQ, UK

[4]International Joint Laboratory for Regional Pollution Control, 52425 Jülich, Germany, and Beijing 100871, China

[5]Collaborative Innovation Center of Atmospheric Environment and Equipment Technology, Nanjing University of Information Science and Technology, Nanjing 210044, China

15  [6]National Centre for Atmospheric Science, School of Earth and Environmental Sciences, The University of Manchester, Manchester, M13 9PL, UK

[#] Currently at Exeter Climate Systems, University of Exeter, Exeter, EX4 4QE, UK

[*]*Correspondence to*: Gordon McFiggans (g.mcfiggans@manchester.ac.uk)



## Abstract.

The phase behaviour of aerosol particles plays a profound role in atmospheric physicochemical processes, influencing their physical and optical properties and further impacting climate and air quality. However, understanding of aerosol phase behaviour is still incomplete, especially that of multicomponent particles which contain inorganic compounds and secondary organic aerosol (SOA) from mixed volatile organic compound (VOC) precursors. We report measurements conducted in the Manchester Aerosol Chamber (MAC) to investigate the aerosol rebounding tendency, measured as "bounce fraction", as a surrogate of particle phase behaviour during SOA formation from photo-oxidation of biogenic (α-pinene, isoprene) and anthropogenic (o-cresol) VOCs and their binary mixtures on deliquescent ammonium sulphate seed.

Aerosol phase behaviour is RH and chemical composition dependent. Liquid (bounce fraction, BF < 0.2) at RH > 80% and non-liquid behaviour (BF > 0.8) at RH < 30% were observed, with a liquid-to-nonliquid transition with decreasing RH between 30%~80%. This RH-dependent phase behaviour ($RH_{BF=0.2, 0.5, 0.8}$) increased towards a maximum with increasing organic-inorganic-mass ratio ($MR_{org/inorg}$) during SOA formation evolution in all investigated VOC systems. With the use of comparable initial ammonium sulphate seed concentration, the SOA production rate of the VOC systems determines the $MR_{org/inorg}$, and consequently the change of the phase behaviour. Although less important than RH and $MR_{org/inorg}$, the SOA composition plays a second-order role, with differences in liquid-to-nonliquid transition at moderate $MR_{org/inorg}$ of ~1 observed between biogenic and anthropogenic-containing VOC systems. The real atmospheric consequences of our results are that any processes changing ambient RH or $MR_{org/inorg}$ will influence their particle phase behaviour. Where abundant anthropogenic VOCs contribute to SOA, compositional changes of SOA may influence phase behaviour at moderate organic mass fraction (~50%) compared with purely biogenic SOA. Further studies are needed on more complex and realistic atmospheric mixtures.



## 1 Introduction

Aerosol particles are ubiquitous in the atmosphere and can act as reaction vessels where physicochemical processes occur. As one of the key physical properties of aerosol particles, the aerosol phase behaviour can significantly impact those physicochemical processes (Martin, 2000). Following the pioneering work of Virtanen et al. (2010), there has been considerable recent efforts to resolve the influences of particle phase behaviour from a number of perspectives mainly relating to retardation of diffusion or mobile components, including water. The viscous solid particles have potential impacts on physicochemical processes, such as constraining gas-particle partitioning of semi-volatile organic species (Vaden et al., 2011;Shiraiwa et al., 2011;Shiraiwa and Seinfeld, 2012;Zaveri et al., 2014;Renbaum-Wolff et al., 2013), heterogeneous reactions or liquid phase reactions (Shiraiwa et al., 2011;Koop et al., 2011;Kuwata and Martin, 2012;Zhang et al., 2018;Martin, 2000). These processes affect secondary organic and inorganic particulate matters formation in the atmosphere, further impacting their optical properties and air quality. Moreover, the ice nucleation abilities in the upper tropospheric conditions and cloud condensation nuclei activation of aerosol particles are affected by the phase behaviour (Pöschl, 2011;Murray et al., 2010;Murray, 2008;Reid et al., 2018;Shiraiwa et al., 2017;Ignatius et al., 2016;Slade et al., 2017), further impacting cloud formation and regional climate. A better understanding the phase behaviour of atmospheric particles is important for understanding physicochemical processes in the atmosphere, and aerosol-cloud interactions.

The particle rebounding property has been widely used to study phase behaviour of aerosol particles (Dahneke, 1971;Stein et al., 1994). In a real atmosphere, the phase behaviour of aerosol particles varied significantly under various environments, depending on the ambient relative humidity (RH) and aerosol chemical composition. For example, background atmospheric particles in the tropical rainforest over central Amazonia, mainly composed of isoprene-derived secondary organic aerosol (SOA), were initially liquid for ambient RH > 80% and temperature of 23-27 °C (Bateman et al., 2015b). In contrast, when the measurement site was influenced by anthropogenic air mass from urban pollution and biomass burning, non-liquid PM fraction increased to 60% at 95% RH (Bateman et al., 2017). However, in the boreal forest of northern Finland, atmospheric particles (mainly monoterpene-derived SOA) showed amorphous, solid-





like phase state (Virtanen et al., 2010). An enhanced fraction of particulate sulphate can lead to loss of particle bounce and atmospheric particles with high fraction of inorganic compounds showed a liquid-
like phase state under moderate and high ambient RH (e.g. urban area (Liu et al., 2017), subtropical coastal megacity (Liu et al., 2019), south-eastern US rural site (Pajunoja et al., 2016), North-eastern near forest area (Slade et al., 2019)). When the terpene-dominant SOA increased during night time (Slade et al., 2019) or ambient RH dropped under 60% (Liu et al., 2017), the non-liquid PM increased.

Consistent with the main findings in the field studies, the particle bounce of pure SOA from the oxidation
of representative biogenic or anthropogenic VOC (e.g. isoprene, α-pinene, toluene) decreased with elevated RH and varied with SOA composition (Bateman et al., 2015a;Saukko et al., 2012). The discrepancy between the Bateman et al. (2015a) and Saukko et al. (2012) was the RH at which aerosol particles fully adhered to the substrate (with < 20% aerosol particles rebounding; referred as the adhesion RH). In a well-calibrated rebounding impactor system, the rebound or adhering behaviour is highly related
to the aerosol phase state, which is determined by material softening. Bateman et al. (2015a) found the complete adhesion RH for the isoprene SOA, α-pinene SOA, 2:1 isoprene/α-pinene mixture and toluene SOA were >65%, >95%, >80% and >80%. However, Saukko et al. (2012) observed that SOA from isoprene and α-pinene oxidation, do not fully adhere to the substrate even RH up to 90%. This discrepancy might due to the slightly different instrumentation design (pressure drop in Saukko et al. (2012) vs
atmospheric pressure in Bateman et al. (2015a)) or SOA composition difference from different oxidation conditions. As observed in the real atmosphere, aerosol particles are usually a mixture with organic and inorganic species (e.g. sulphate and nitrate) (Jimenez et al., 2009). Since the inorganic species have lower glass transition temperature than atmospheric-relevant organics (Pedernera, 2008;Koop et al., 2011), the presence of inorganic species could theoretically lower the glass transition temperature of the estimated
SOA compounds mixtures. Furthermore, inorganic species are hydrophilic and the absorbed water molecules can act as a plasticiser, which effectively lower the glass transition temperature and soften the aerosol (Koop et al., 2011;Martin, 2000). Therefore, ignoring the mixing with inorganic species, or assuming external mixing as in Shiraiwa et al. (2017), could bias our understanding on the phase behaviour of aerosols containing abundant inorganic compounds.





There are few studies on phase behaviour of multicomponent aerosol particles. Saukko et al. (2012) found increasing sulphate fraction mixing with SOA produced by longifolene oxidation can reduce the particle rebounding significantly. Saukko et al. (2015) extended this to study deliquescence hysteresis of ammonium sulphate with condensed SOA from α-pinene and isoprene oxidation as manifested by particle rebound. They found the α-pinene SOA condensing on ammonium sulphate seed particles showed no
influences on their deliquescence but significantly attenuated the efflorescence behaviour. In contrast, the isoprene SOA system resulted in a loss of sharp deliquescence and efflorescence behaviour in comparison to pure ammonium sulphate (Saukko et al., 2015). These contrasting findings above indicate that aerosol phase behaviour could differ between different SOA precursor systems as well as in the presence of inorganic compounds. However, it is still unclear how the phase behaviour (and any associated diffusion
limiting behaviour) will be influenced during the formation and evolution of SOA from an increased complexity of mixed precursors in the presence of inorganic seed.

We designed a series of aerosol simulation chamber experiments to study SOA formed from representative biogenic (α-pinene, isoprene) and anthropogenic (*o*-cresol) VOC photochemistry. The experiments studied single VOC precursors and their binary mixtures under modest-$NO_x$ condition on
deliquescent ammonium sulphate seed particles. We frame our results around the following hypotheses:

1) Aerosol phase behaviour for SOA mixture is driven by RH and organic-inorganic-mass ratio

2) The difference in SOA composition is less important in determining the phase behaviour than the SOA production rate, which changes the organic-inorganic ratio.

The main objective of this paper is to test the above two hypotheses and discuss their potential
atmospheric implications.



## 2 Measurements and Methods

### 2.1 Reaction chamber and experimental setup

The aerosol particles for the experiments were produced in the Manchester Aerosol Chamber (MAC). The facility is run as a batch reactor with an 18 $m^3$ (3m (H)*3m (L)*2m (W)) FEP Teflon bag supported by three aluminium frames, in which the upper and the lower frame can move freely with the expansion or collapsing as sampling air is introduced to or extracted from the chamber. The Teflon bag is enclosed inside a housing with temperature and relative humidity controlled by an air conditioning system. Two 6kW Xenon arc lamps (XBO 6000 W/HSLA OFR, Osram) and a series of halogen bulbs arranged in 7 rows containing 16 bulb each (Solux 50W/4700K, Solux MR16, USA) are mounted inside of the enclosure housing. The combination of 5 rows of halogen bulbs (2 rows spare) and two Xenon arc lamps is chosen to be a good representative to mimic the solar spectrum in the wavelength of 290-800 nm (Alfarra et al., 2012). The calculated photolysis rate of $NO_2$ ($j_{NO2}$) from $O_3$-NO-$NO_2$ photostationary state was 1.8-3 $\times$ $10^{-3}$ $s^{-1}$ during the experiment period. The simulated irradiation spectrum in our chamber is comparable to the measured solar spectrum at the mid-day on a clear sky day in June of Manchester (https://www.eurochamp.org/Facilities/SimulationChambers/MAC-MICC.aspx).

To ensure chamber cleanliness and data reproducibility, an automatic fill/flush cycle is conducted pre- and post-experiment using 3 $m^3$ $min^{-1}$ purified clean air. 5~6 cycles are routinely conducted between experiments, with total number concentration of aerosol particles usually lower than 10 p/cc after cleaning. Air is scrubbed using a series of filters, including Purafil (Purafil Inc., USA) and charcoal removing reactive gaseous compounds and HEPA (Donaldson Filtration, USA) removing particles and a dryer. To remove reactive compounds, the chamber is soaked in high concentrations of $O_3$ (~1 ppm) overnight between experiments which is removed during pre-experiment fill/flush cycles on the subsequent day. An additional harsh cleaning procedure is conducted weekly with high $O_3$ (~1 ppm) and UV for 4~5 hours to consume the remaining reactive compounds.

Seed particles, VOC, $NO_x$ and water vapour are injected into chamber before illumination. Deliquescent ammonium sulphate (AS) seed (Puratronic, 99.999% purity) are nebulized and introduced into a drum to


mix before flushing into chamber. Liquid VOC precursors (α-pinene, isoprene, $o$-cresol; Sigma Aldrich, GC grade ≥ 99.99% purity) are injected using a syringe into a heated glass bulb in which the VOCs are instantaneously vaporized. The vaporized VOCs are flushed into the chamber with a flow of ~ 0.5 bar

high purity nitrogen (ECD grade, 99.997%). Here, the experiments will investigate single and iso-reactive binary mixtures under modest-$NO_x$ condition (VOC/$NO_x$ of 5~10). $NO_x$ (mostly as $NO_2$ in this study) is introduced through a cylinder with a flow of ~0.3 bar high purity nitrogen (ECD grade, 99.997%) and the $O_3$-NO-$NO_2$ photostationary state will establish immediately after illumination. The detailed initial conditions for chamber setup are shown in Table 1.

**Table 1. Summary of the initial conditions of chamber experiment.**

| Exp. Date | VOC type | [VOC]$_0$ (ppbV) | VOC/$NO_x$ | T (°C) | RH (%) | AS Seed conc. (µg m$^{-3}$)[a] | O:C ratio |
|---|---|---|---|---|---|---|---|
| 28-Mar-2019 | α-pinene | 309 | 7.7 | 26.7 | 50.5 | 72.6 | 0.36 ± 0.03 |
| 17-Apr-2019 | α-pinene | 155 | 4.4 | 25.9 | 55.0 | 47.8 | 0.45 ± 0.04 |
| 02-Apr-2019 | Isoprene | 164 | 6.8 | 27.2 | 47.3 | 64.1 | n.a. |
| 12-Apr-2019 | $o$-cresol | 400 | n.a. | 27.3 | 53.3 | 47.8 | 0.64 ± 0.03 |
| 19-Apr-2019 | $o$-cresol | 200 | 5.0 | 26.9 | 51.3 | 51.3 | 0.66 ± 0.05 |
| 08-Apr-2019 | α-pinene/isoprene | 237 (155/82) | 9.9 | 27.0 | 48.4 | 62.0 | 0.43 ± 0.05 |
| 23-Apr-2019 | α-pinene/$o$-cresol | 355 (155/200) | n.a. | 25.6 | 55.8 | 42.5 | 0.48 ± 0.05 |
| 18-Apr-2019 | $o$-cresol/isoprene | 282 (200/82) | 8.3 | 27.1 | 52.7 | 49.6 | 0.69 ± 0.05 |

[a] calculated mass concentration from volume concentration from DMPS with a density of 1.77 g/cm$^3$.



## 2.2 Instrumentation

A series of instruments are equipped for gas-phase and particle-phase measurements in MAC. NO, $NO_2$

and $NO_x$ are recorded by NO-$NO_2$-$NO_x$ analyser (Model 42i, Thermo, USA) and $O_3$ is measured by $O_3$ analyser (Model 49C, Thermo, USA). RH and T inside of MAC are recorded by an Edgetech dewpoint hygrometer (DM-C1-DS2-MH-13, USA) and two Sensirion SHT75 sensors (Farnell 413-0698, USA). The aerosol number size distribution (20-550 nm) is measured by a Differential Mobility Particle Sizer (DMPS).

The particle bounce behaviour (bounce fraction, BF) was measured by a three-arm particle rebound apparatus with RH adjustment system. A brief instrumental description is provided below and more details can be found in Bateman et al. (2014) and Liu et al. (2017) along with a schematic diagram in Figure S1 of the latter. Three single-stage impactors operated in parallel in the system combined with a condensation particle counter (CPC, Model: 3772, TSI, USA). The first impactor is not equipped with a

plate (step 1 in Figure S1 in Liu et al. (2017)), so particles can pass through the first impactor directly, measuring the total particle population ($N_1$). The second impactor is equipped with a smooth plate (step 2 in Figure S1 in Liu et al. (2017)), which provides a solid surface and allows particles rebounding from the impactor. The particle population measured after the second impactor represents the sum of particles that don't strike the impactor and that strike but rebound from the impactor ($N_2$). The third impactor is

equipped with a grease-coated plate (step 3 in Figure S1 in Liu et al. (2017)). The coated grease is quite sticky and all particles striking on the plate will be stuck. Therefore, the particle number population after grease-coated plate provides a measure of the particles that don't strike the impactor ($N_3$). The rebound fraction BF is defined in equation [1].

$$BF = \frac{N_2 - N_3}{N_1 - N_3} \qquad [1]$$

BF can be used as a proxy of aerosol phase behaviour, with limited capability representing semi-solid or solid particles over 100 Pa s in volatility (Bateman et al., 2015a;Reid et al., 2018). Nevertheless, it





provides insights into the transition process between liquid and solid/semisolid phase, referred to as liquid-to-nonliquid transition below.

The chemical composition of the non-refractory PM$_1$ components (NH$_4^+$, NO$_3^-$, SO$_4^{2-}$, SOA) was recorded by a High-Resolution Time-of-Flight Aerosol Mass Spectrometer (HR-ToF-AMS, Aerodyne Research Inc., USA). A detailed introduction of the instrument, calibration procedures and quantification of the aerosol concentrations were described previously (DeCarlo et al., 2006;Jayne et al., 2000;Allan et al., 2003;Allan et al., 2004). The instrument was operated in 'V mode', and recorded with a time resolution of 1 min (30 s mass spectrum (MS) + 30 s particle-time-of-flight (PToF)).

Calibrations were performed before and after the campaign using monodisperse (350 nm) ammonium nitrate and ammonium sulphate particles following the standard procedure in Jayne et al. (2000) and Jimenez et al. (2003). An averaged ionization efficiency of nitrate was $9.38 \times 10^8$ from the two calibrations. According to the ion balance of ammonium nitrate and ammonium sulphate in the calibrations, the specific relative ionization efficiencies (RIE) for NH$_4^+$ and SO$_4^{2-}$ are determined as $3.57 \pm 0.02$ and $1.28 \pm 0.01$, respectively. The RIE of all organic compounds used the default value of 1.4 (Alfarra et al., 2004). In this study, real-time collection efficiency was applied to the data analysis by comparing NR-PM$_1$ mass with DMPS mass.

## 2.3 Rationale behind the choice of precursor

The real atmosphere comprises a complex mixture of VOCs with various reactivities, many of which may act as SOA precursors with varying degrees of efficiency. Recent chamber studies have started using relatively simple VOC mixtures to investigate complex interactions in their ability to form SOA in the presence of inorganic seed particles (McFiggans et al., 2019;Shilling et al., 2019). Extending these previous studies, a project was designed aiming at characterising chemical mechanisms, yield, and physicochemical properties (volatility, hygroscopicity, CCN activity, phase behaviour) of SOA formed from iso-reactive biogenic/anthropogenic VOC photochemistry on ammonium sulphate seed and exploring potential implications to the real atmosphere. Building on the McFiggans et al. (2019), we added a representative anthropogenic VOC (*o*-cresol) into initially designed binary mixtures of biogenic



VOCs (isoprene & α-pinene). *o*-cresol is chosen by virtue of its ·OH reactivity comparable with the chosen biogenic VOCs (Coeur-Tourneur et al., 2006), which enables comparable initial concentration to

have an equal reactivity with ·OH at the beginning of experiment (referred as iso-reactive). The modest SOA yield of *o*-cresol (Henry et al., 2008) gives a good contrast with the low-yield isoprene and high-yield α-pinene. The overall experiment design enables us to explore SOA formation in iso-reactive single, binary and ternary VOC mixtures oxidation. This manuscript focuses on aerosol phase behaviour of seeded SOA from iso-reactive single and binary VOCs photochemistry. Unfortunately, the bounce

impactor was unavailable for the ternary mixture experiment.

## 3 Results and Discussion

### 3.1 BF dependence on RH and organic-inorganic-mass ratio

Figure 1 shows the rebound curves of the 100~200 nm multicomponent aerosol particles formed in various VOC systems for the period when organic mass fraction in NR-PM$_1$ is larger than 0.05. For all

investigated VOC systems, the aerosol particles exhibited BF > 0.8 at RH < 30% and BF < 0.2 at RH > 80% at room temperature (18 °C). Between 30% and 80% RH, the BF monotonically decreased with the increasing RH, indicating a gradual transition in BF (usually within 15~25% RH width for BF declining from > 0.8 to < 0.2). Assuming the aerosol particles to be non-liquid if their BF > 0.8 and liquid if BF < 0.2 (Bateman et al., 2015a;Liu et al., 2017), this implies a gradual transition with RH in all investigated

VOC systems in contrast to the rapid dissolution corresponding to deliquescence of inorganic salt particles (Tang and Munkelwitz, 1993;Kreidenweis and Asa-Awuku, 2014). Owing to limitations of the technique in differentiating particle viscosity at high values (Bateman et al., 2014), the non-liquid phase could represent semi-solid or solid phase. In addition, the rebound curves varied along with SOA formation and subsequent evolution as the photochemistry continues in all the investigated VOC systems (as shown in

Figure 1). To illustrate the influences of chemical composition, the overview of rebound curves as a function of organic-inorganic-mass ratio (MR$_{org/inorg}$) in all VOC systems is shown in Figure S1, indicating the potential important role of MR$_{org/inorg}$ (and maybe OA composition) in determining the phase behaviour as RH.





To clearer describe the phase behaviour during SOA formation evolution, the $RH_{BF=0.2}$, $RH_{BF=0.5}$, $RH_{BF=0.8}$

are determined representing the RH at which the BF is close to 0.2 (± 0.15), 0.5 (± 0.15), 0.8 (± 0.15). It is worth noting that the determination of $RH_{BF=0.2, 0.5, 0.8}$ carry a maximum error of ± 5% owing to the measurement resolution and resultant number of data points. By tracking the variation of $RH_{BF=0.2, 0.5, 0.8}$ during SOA formation evolution, we can understand the tendency of phase behaviour change and insight into the liquid-to-nonliquid transition at the variation of RH at which BF changes from 0.8 to 0.2 among

VOC systems. As different VOCs have different reactivity with oxidant and yield, the formed SOA mass from different VOCs after six-hour photo-oxidation, and consequently the $MR_{org/inorg}$, are different among VOC systems. For *o*-cresol/isoprene, 50% reactivity *o*-cresol, *o*-cresol, α-pinene/isoprene, α-pinene/*o*-cresol, 50% reactivity α-pinene and α-pinene systems, the $MR_{org/inorg}$ reached up to 0.53 ± 0.02, 1.27 ± 0.05, 2.47 ± 0.06, 3.77 ± 0.11, 3.77 ± 0.01, 4.45 ± 0.04, 7.37 ± 0.06 respectively, for the last rebound

curve at the end of experiments. As shown in Figure 2, the $RH_{BF=0.2, 0.5, 0.8}$ increased towards a maximum as an increase of the $MR_{org/inorg}$ in all investigated VOC systems, indicating an increase of rebound tendency during SOA evolution on deliquescent sulphate seed. Figure 2a-c shows a co-increasing trend between $RH_{BF=0.2, 0.5, 0.8}$ and $MR_{org/inorg}$ as $MR_{org/inorg} < 2$, thereafter $RH_{BF=0.2, 0.5, 0.8}$ remains constant at 70~75%, 65~70% and 50~65%, respectively, in all investigated VOC systems. Herein, the reason for the

decrease of $RH_{BF=0.5}$ from 60% to 50% at $MR_{org/inorg}$ ~1 in the *o*-cresol system is not known and the same behaviour was not observed for $RH_{BF=0.2, 0.8}$.

As expected, the change in $MR_{org/inorg}$ during SOA formation and evolution differed in various VOC systems (as shown in Figure 3), depending on their SOA production rate (as shown in Figure S2), noting the comparable initial sulphate seed concentrations. The order of the SOA production rate (from low to

high) in all the investigated VOC systems are, o-cresol/isoprene, 50% reactivity *o*-cresol, *o*-cresol, α-pinene/isoprene, α-pinene/*o*-cresol, 50% reactivity α-pinene and α-pinene (note that insufficient mass was generated from isoprene in the presence of a neutral ammonium sulphate seed). It is worth noting that the SOA production rate in α-pinene/isoprene, α-pinene/*o*-cresol and 50% reactivity α-pinene systems are very similar, and their order is determined by the minor difference of the slopes in Figure S2. Clearly, a

higher SOA production rate increases the $MR_{org/inorg}$ faster and consequently the phase behaviour change. As shown in Figure 4, $RH_{BF=0.2, 0.5, 0.8}$ showed an increasing trend towards a maximum across the various



VOC systems, highly related to how fast the SOA are formed. With an increasing SOA formation rate from the lowest $o$-cresol/isoprene to highest α-pinene system, a shorter time was taken for $RH_{BF=0.2, 0.5, 0.8}$ to reach the maximum. Take $RH_{BF=0.5}$ as an example, it can be seen from Figure 1 that the $RH_{BF=0.5}$ at the

beginning of the photochemistry is lower than 40% for all investigated VOC systems. Panel a) shows that, for the $o$-cresol/isoprene system with the lowest SOA formation rate, it took ~3.5h for $RH_{BF=0.5}$ to increase to 50% and ~5h to approach 60%. For comparison, for α-pinene/isoprene (the moderate) and α-pinene (the highest) systems, it takes only ~1.5h and 0.5h for the $RH_{BF=0.5}$ increasing to 50% and 70%, respectively. Similar results can also be found for the cases in $RH_{BF=0.2, 0.8}$.

It can be seen that in all of the investigated VOC systems, when the SOA mass fraction > 0.05, the phase dependence on $MR_{org/inorg}$ was qualitatively similar irrespective of the single/binary biogenic/anthropogenic SOA precursors. That is, the $RH_{BF=0.2, 0.5, 0.8}$ increased towards a maximum value with an increase of the $MR_{org/inorg}$ during SOA formation evolution. In regard to the pure SOA from biogenic or anthropogenic VOCs oxidation showed amorphous solid property rather than the expected

liquid phase as deliquescent inorganic particles (Bateman et al., 2015a;Virtanen et al., 2010;Saukko et al., 2015;Saukko et al., 2012), such as the $RH_{BF=0.2, 0.5, 0.8}$ of pure SOA formed from α-pinene and toluene photo-oxidation were 80~90%, 80~85%, and ~65%, respectively (Bateman et al., 2015a). It is expected that the increasing aerosol rebounding tendency can happen with increasing SOA mass condensing on the deliquescent sulphate seed. This speculation is supported by our results that an increase of $RH_{BF=0.2, 0.5, 0.8}$

toward the maximum with more SOA condensation. Additionally, the maximum $RH_{BF=0.2, 0.5, 0.8}$ at high $MR_{org/inorg}$ up to ~8 in our study were 10~15% lower than pure SOA mentioned above, indicating the presence of small mass fraction of inorganic compounds (~10%) makes multicomponent aerosol particles bounce less than pure SOA. Moreover, the time taken for $RH_{BF=0.2, 0.5, 0.8}$ to reach the maximum depends on how rapidly the SOA was formed, shorter time for the faster SOA production rate of the investigated

VOC systems. This general behaviour was independent of the yield of the VOC and whether the precursor was biogenic or anthropogenic. In addition, the individual VOC systems behaved in the same way as the VOC mixtures.





As shown in Figure 3, it should be noted that, to avoid high noise-signal-ratio for low organic mass loading measured by HR-ToF-AMS, the data points with organic mass fraction larger than 0.05 (assuming

uniform chemical composition) were selected for consideration. It was observed that particulate nitrate (Max. 5% ~ 16% of NR-PM$_1$) is formed during photochemistry in all investigated VOC systems. Nitrate can be formed either from oxidation of NO$_2$ followed by neutralisation by NH$_3$ or organic oxidation product (organic nitrate). The organic nitrate fraction in total nitrate signal can be estimated following the NO$^+$/NO$_2^+$ ratio method proposed by Farmer et al. (2010) given the differentiation of NO$^+$/NO$_2^+$ ratio for

pure NH$_4$NO$_3$ (2.5, from calibrations) and organic nitrate of 10~15 (Bruns et al., 2010;Fry et al., 2009;Kiendler-Scharr et al., 2016;Reyes-Villegas et al., 2018). As shown in Figure S3, organic nitrate fraction in total nitrate signal was lower than 15% in almost all investigated VOC systems except for the last two hour of o-cresol system (up to ~20%). This NO$^+$/NO$_2^+$ ratio method can indicate organic nitrate statistically only if the organic nitrate fraction is larger than 15% (Bruns et al., 2010). Thus the observed

particulate nitrate is mainly inorganic nitrate with small contribution of organic nitrate (< 20%) for all VOC systems in this study. The small contribution of organic nitrate mass have little influences on the onset of organic mass fraction > 0.05. Moreover, Li et al. (2017) found that the pure NH$_4$NO$_3$ particles adhered on the impactor even the RH has been reduced to ~ 5%. It is worth noting that the variable particulate nitrate across all VOC systems in this study might have some influence on the phase behaviour

of the multicomponent aerosol particles during SOA formation evolution and the influence of changing inorganic component ratios on the BF in multicomponent mixtures should be the focus of this work.

## 3.2 BF dependence on OA composition

In addition to the control of the BF by MR$_{org/inorg}$, there is an indication that the SOA composition across the various VOC systems may also impact aerosol phase behaviour. As indicated in Figure 2 b-c, the

increase of the RH$_{BF=0.5, 0.8}$ as a function of MR$_{org/inorg}$ is less rapid in the α-pinene/isoprene, 50% reactivity α-pinene and α-pinene system (referred as BVOC systems) than the anthropogenic-containing VOC systems (AVOC-containing systems, including o-cresol, 50% reactivity o-cresol, o-cresol/isoprene, α-pinene/o-cresol). It can been seen that, for the AVOC-containing systems, the RH$_{BF=0.5}$ and RH$_{BF=0.8}$ were 65~70% and 40~55% when MR$_{org/inorg}$ approached ~1, whereas only 40~45% and ~25% in the 50%





reactivity α-pinene and α-pinene/isoprene systems (BVOC systems). Interestingly, with more SOA condensing, the discrepancy disappeared and the $RH_{BF=0.5}$ and $RH_{BF=0.8}$ converged for BVOC and AVOC-containing systems as $MR_{org/inorg} >2$. In contrast, the $RH_{BF=0.2}$ as a function of $MR_{org/inorg}$ was the same for BVOC and AVOC-containing mixtures. This indicates the decreased RH to achieve liquid-to-nonliquid transition for BVOC and AVOC-containing systems was different at moderate $MR_{org/inorg}$ of ~1 but

quantitatively similar at low and high $MR_{org/inorg}$.

The above evidence indicates VOC type (BVOC or AVOC-containing), thereafter SOA composition, can play a second-order role in the phase behaviour of multicomponent aerosol particles. Here, we use degree of oxidation of the SOA (O:C atomic ratio derived from HR-ToF-AMS) as a proxy of SOA composition to explore its relation with phase behaviour. As shown in Table 1, for BVOC systems, the averaged atomic

O:C ratio of SOA in α-pinene/isoprene, 50% reactivity α-pinene, and α-pinene systems, are $0.43 \pm 0.05$, $0.45 \pm 0.04$, $0.36 \pm 0.03$, respectively. For AVOC-containing systems, the O:C ratio of SOA was 10~50% higher than BVOC systems, with $0.69 \pm 0.05$, $0.66 \pm 0.05$, $0.64 \pm 0.03$, $0.48 \pm 0.05$ in *o*-cresol/isoprene, 50% reactivity *o*-cresol, *o*-cresol and α-pinene/*o*-cresol systems, respectively. In addition, the $RH_{BF=0.2, 0.5, 0.8}$ response to the $MR_{org/inorg}$ is coloured according to O:C ratio in Figure 2. It can be seen that the O:C

ratio is characteristic for the SOA precursors and shows little variation through individual experiments. No direct relationship between O:C ratio and the phase behaviour change was observed during SOA formation evolution among all investigated VOC systems. Previous studies have shown a tentative dependence on the O:C ratio of the organic particles with phase behaviour (e.g. glass transition temperature) (Koop et al., 2011;Shiraiwa et al., 2017). In contrast, Saukko et al. (2012) showed the O:C

ratio only influence rebounding behaviour of SOA formed from *n*-heptadecane in a Potential Aerosol Mass (PAM) reactor, with no influence in the α-pinene, longifolene, and naphthalene systems. There is some evidence showing that aerosol morphology (single well-mixed phase or phase separation) is closely related to O:C ratio of SOA and organic-inorganic-mass ratio (Bertram et al., 2011;Krieger et al., 2012;You et al., 2014;Freedman, 2017;Song et al., 2012;Smith et al., 2013). It is unknown whether the

morphology plays a role in the phase behaviour discrepancy between BVOC and AVOC-containing systems at moderate $MR_{org/inorg}$ considering their O:C difference, which is of interest and need further investigations.



## 4 Conclusions and implications

Our experiments support the validity of our two proposed hypotheses: 1) Aerosol phase behaviour for
SOA mixture is determined by RH and organic-inorganic-mass ratio ($MR_{org/inorg}$). 2) The difference in
SOA composition is less important in determining the phase behaviour than the rate of SOA production,
which changes the $MR_{org/inorg}$.

First, aerosol phase behaviour is clearly RH-dependent as already widely known. Multicomponent aerosol
particles were always found to exhibit liquid-like behaviour (BF < 0.2) above 80% RH, and nonliquid-
like behaviour (BF > 0.8) at below 30% RH. The bounce measurements always indicated continuously
increasing nonliquid-like behaviour as RH decreased from 80% to 30%. These RH-dependent rebound
curves strongly depend on the increase in $MR_{org/inorg}$ during SOA formation in all investigated VOC
systems. The identified $RH_{BF=0.2, 0.5, 0.8}$ increased towards a maximum with the increase of the $MR_{org/inorg}$,
and the increase rate of $RH_{BF=0.2, 0.5, 0.8}$ is determined by the SOA production rate on sulphate seed across
the VOC systems. This general behaviour was independent of the yield of the SOA precursors and
whether the precursor was biogenic or anthropogenic. In some ways, this is an obvious result that follows
from the rate of SOA mass increase in the system under investigation. However, this production rate will
be dependent on the reactivity and yield of VOCs in the mixture, their concentrations and interactions
influencing SOA mass formation. This set of complex dependencies will control the changes in particle
$MR_{org/inorg}$ in mixtures and hence in phase behaviour.

Although less important than the RH and $MR_{org/inorg}$, the SOA composition plays a secondary role
affecting the phase behaviour. It was observed that the $RH_{BF=0.2}$ as a function of $MR_{org/inorg}$ was the same
for BVOC and AVOC-containing systems, however the decreased RH to achieve liquid-to-nonliquid
transition (BF from ~0.2 to ~0.8) was different at moderate $MR_{org/inorg}$ of ~1 but quantitatively similar at
low and high $MR_{org/inorg}$. For example, at moderate $MR_{org/inorg}$ of ~1, the $RH_{BF=0.2}$ for BVOC and AVOC-
containing systems was 65-70%. To achieve liquid-to-nonliquid transition with BF of ~0.8, the RH
needed to decrease to 40~55% in AVOC-containing systems but to ~25% in BVOC systems. This
discrepancy cannot be explained by the atomic O:C ratio of SOA during SOA formation evolution. It

should be the focus of future work to investigate whether SOA composition has a more pronounced effect

outside the dynamic O:C range of the mixtures in our experiments.

Multicomponent aerosol phase behaviour depends on RH and organic-inorganic-mass ratio in a particle and in the atmosphere as well as our chamber. Any interactions influencing these two factors will therefore influence the phase behaviour. For example, a fast formation of SOA or significant ambient RH change in real atmosphere could change phase behaviour of particles, and consequently influencing

atmospheric physicochemical processes. There is an additional indication that an increased anthropogenic VOC contribution to the mixture could give different phase behaviour at certain moderate organic mass fraction (~50%), hence there will be some second-order differences depending on the relative contributions of anthropogenic and biogenic VOC. Further studies should be carried out on more complex and realistic atmospheric mixtures.

## 380 **Data availability**

The observational dataset of this study is available on the open dataset of EUROCHAMP-2020 programme (https://data.eurochamp.org/data-access/chamber-experiments/).

## **Author contributions**

G.M., M.R.A., Y.W., A.V. and Y.S. conceived the study. G.M. and Z.W. co-applied Trans-National

Access (TNA) funding from EUROCHAMP for the phase behaviour measurement. Y.W., A.V., Y.S. and D.M. conducted the chamber experiments and collected the dataset. T.Z. and X.M. operated the rebound impactor apparatus. Y.C. and D.H. provided helpful discussions. Y.W. performed data integration, data analysis and wrote the manuscript with the inputs from all co-authors under the guidance of G.M. and M.R.A.



## Acknowledgement

Z.W. acknowledges National Natural Science Foundation of Chin (41875149). Y.W. acknowledges the joint-scholarship of The University of Manchester and Chinese Scholarship Council. M.R.A. acknowledges support by UK National Centre for Atmospheric Sciences (NACS) funding. A.V. acknowledges the Natural Environment Research Council (NERC) EAO Doctoral Training Partnership funding. This work is supported by the following projects: Trans-National Access (TNA) of EUROCHAMP-2020. We acknowledges AMF/AMOF for providing SMPS instrument (AMF_25072016114543 and AMF_04012017142558).

## Competing interests

All authors declare no conflict of interest.

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

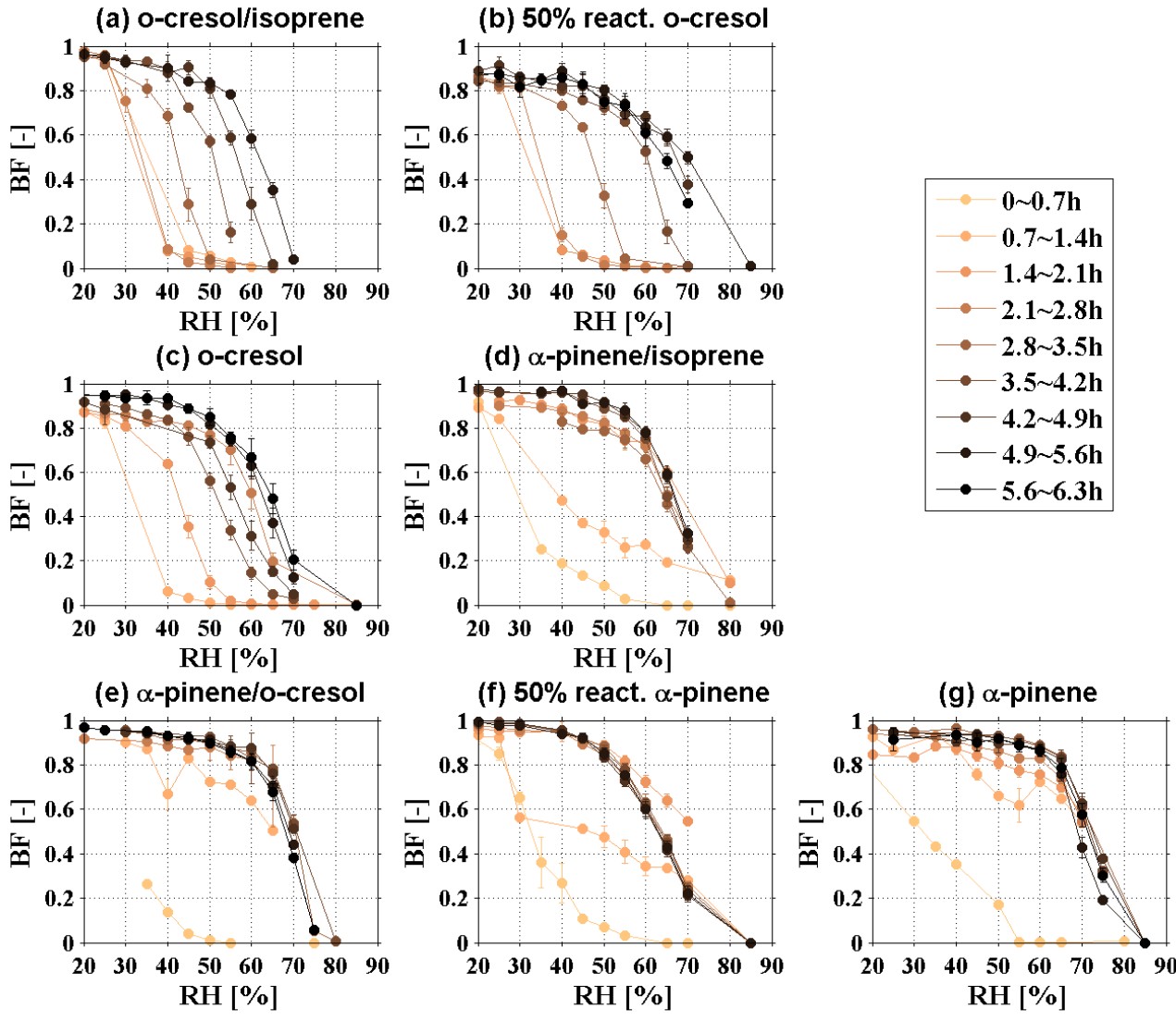

Figure 1. Time series of the Bounce fraction (BF) as a function of RH measured by three-arm particle rebound apparatus with RH adjustment system of the 100-200 nm secondary organic aerosol (SOA) formed from various iso-reactive single/binary biogenic/anthropogenic VOCs (volatile organic compounds) photochemistry under relative low-$NO_x$ condition on deliquescent ammonium sulphate seed (as the SOA mass fraction in NR-$PM_1$ is larger than 0.05).




Figure 2. The measured (a) $RH_{BF=0.2}$, (b) $RH_{BF=0.5}$ and (c) $RH_{BF=0.8}$ as a function of organic-inorganic ratio ($MR_{org/inorg}$) and coloured by atomic O:C ratio in various VOC systems photochemistry on deliquescent ammonium sulphate seed. Black box points out the $RH_{BF=0.2,\ 0.5,\ 0.8}$ at moderate $MR_{org/inorg}$ of ~1 for biogenic VOC systems (α-pinene/isoprene, 50% reactivity α-pinene).





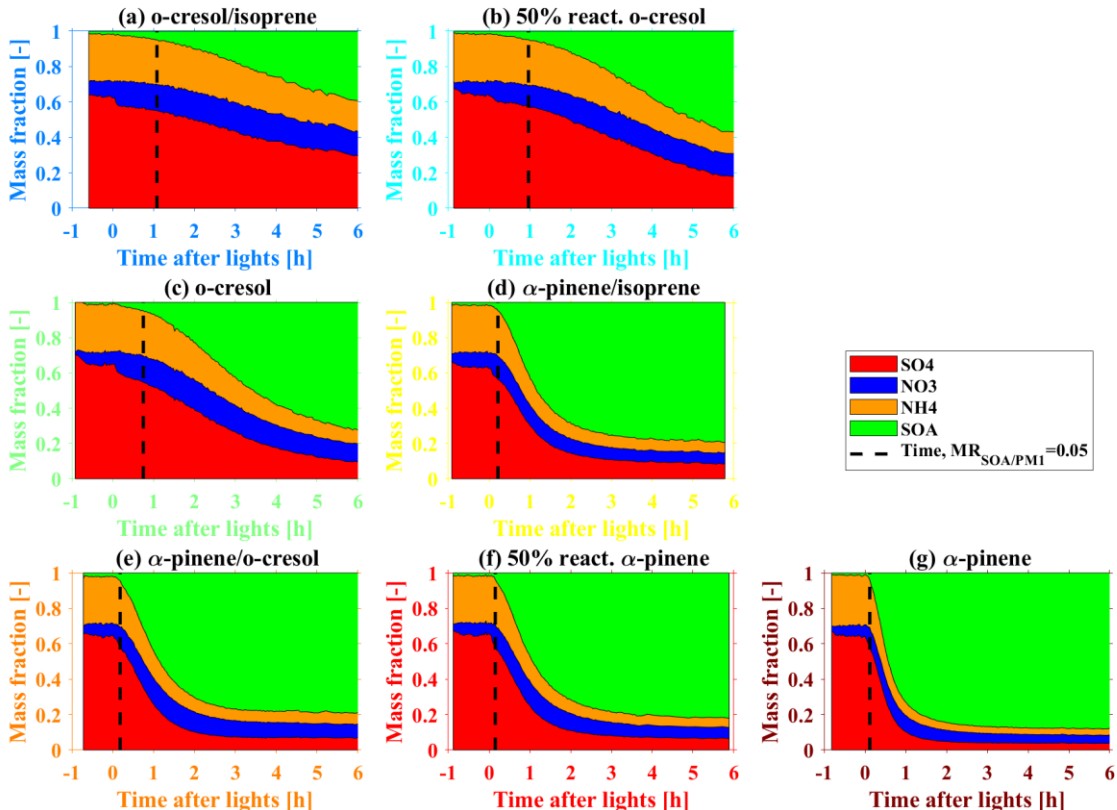

Figure 3. Summary of mass fraction of chemical species (SO4, NO3, NH4, SOA) in bulk NR-PM$_1$ measured by HR-ToF-AMS in various VOC systems photochemistry on deliquescent ammonium sulphate seed. The black dashed line represents the defined time where SOA mass fraction is 0.05.




Figure 4. The time series of measured (a) $RH_{BF=0.2}$, (b) $RH_{BF=0.5}$ and (c) $RH_{BF=0.8}$ of the multi-component aerosol particles in various VOC systems photochemistry on deliquescent ammonium sulphate seed.