# Peer review of "Secondary organic aerosol phase behaviour in chamber photo-oxidation of mixed precursors"

_Atmospheric Chemistry and Physics, 2021_

## Author Comment (AC1)

**Response to comments of anonymous referees # 1**

**General comments**

The authors presented laboratory results of phase behaviour of secondary organic aerosol (SOA) mixed with ammonium sulphate seed particles using smog-chamber experiments and particle bouncing measurements. The key novelty about this study is the usage of "isoreactive" (see below for a question on this) single- or mixed-precursor volatile organic compounds (VOCs) from both biogenic and anthropogenic sources, while the two hypotheses (RH rules and MR_org/inorg dominates) tested are actually well established. The results confirmed the vital role of RH and the dominating effect of the fraction of organics in the aerosol particles in the changes of phase behavior, while also suggested a second-order role of SOA composition (whether it is anthropogenic or biogenic, but not very much related to O:C ratio) when MR_org/inorg is close to unity. The experiments of this study were carefully designed and data well interpreted, in support of the conclusions made. The manuscript is also fairly well written. I have a few comments for the authors to clarify, and recommend a Minor Revision before publication.

*Many thanks to the reviewer for the comments and suggestions. We have improved the manuscript accordingly. Please find a point-by-point response below.*

**Specific comments:**

**Q1.** "iso-reactive": I assume that this term is referring to a similar reactivity towards a certain oxidant for the VOCs or mixtures chosen (Table 1). The question is, what is the dominating oxidant in the chamber experiments? OH radicals or O3? Or a mixture of them? If it is the mixture, is it a combined reactivity that makes them "iso-reactive"? Please clarify.

*Good point. Yes, the term "iso-reactive" in this study means that the initial concentration of VOCs chosen in each system were adjusted towards the OH reactivity, to ensure each VOCs have the equal chance to react with OH radical at the beginning of the experiments. But after the photochemistry starts and the ozone builds up, it is not iso-reactive furthermore. Both OH radical and the formed ozone can oxidize α-pinene and isoprene (OH is the dominant oxidant), but OH radical is the only oxidant for o-cresol oxidation. Here, we take the α-pinene experiment as an example as it forms the maximum ozone concentration (peak: 25 ppb) among all systems. The lifetime of α-pinene towards OH and ozone at the 25-ppb ozone is 5.6 h and 6.4 h. In other systems, the ozone concentration is much lower than 25 ppb so the dominant oxidant for α-pinene and isoprene is OH radical.*

*To avoid repetition, I added a sentence in Sec. 2.1 clarifying that the experimental design will be described in detail in the Sec. 2.3, and the explanation of the iso-reactive towards the oxidants has been added into the Sec. 2.3 when introduce the rationale behind the choice of precursor and the iso-reactivity.*

*"The initial VOC concentrations were adjusted toward OH reactivity (dominant oxidant) to achieve the equal chance to compete the OH radical at the beginning of the experiments (referred to as iso-reactivity). When the photochemistry starts and the $O_3$ builds up, it is not iso-reactive furthermore during the experiments as the formed $O_3$ during the photochemistry can oxidize α-pinene and isoprene but not for o-cresol."*

**Q2.** About AMS results: a) the authors used O:C and H:C ratios and specific ion signals of NO^+ and NO_2^+ in their analysis, but indicated that only V-mode data were available from AMS measurements (L188). How were those data on elemental ratios and specific ion peaks obtained? Approximated from empirical formulas or high-res fitting on V-mode data? b) L192: how is this ionization efficiency defined here? If it is the number of ammonium nitrate molecules ionized per number of ammonium nitrate molecules introduced, it should be a number much lower than 1. c) 196: please specify on what is "real-time" collection efficiency.

*Thanks for your helpful comment. About the AMS, the answers are shown as below:*

*a) The elemental ratio of O:C used for the proxy of organic oxidation state and the ions of NO+ and NO2+ used for the organic nitrate fraction estimation are derived from the high-resolution fitting on V-mode data. This sentence has been added in the method section 2.2 as suggested.*

*b) Yes, the ionization efficiency means the number of ammonium nitrate molecules ionized per number of ammonium nitrate molecules introduced. The number was wrongly written and has been corrected to $9.38 \times 10^{-8}$ ions molecule$^{-1}$.*

*c) This "real-time" collection efficiency is not a precise description. Here, we corrected the AMS mass concentrations by comparing to the real-time DMPS unit mass concentration multiplied by the calculated density from AMS chemical composition. The mass ratio of AMS/(DMPS\*density) is 0.4 ~ 1.0 in our study, which was used to scale up the mass concentrations of AMS accordingly. This correction only changed the absolute mass but not the ratio of SOA/inorganic or SOA/PM$_1$. And the ratio is the focus of this study.*

*The original sentence ('In this study, real-time collection efficiency was applied to the data analysis by comparing NR-PM1 mass with DMPS mass.') has been replaced by the following sentences:*

*"In this study, the AMS mass concentrations were corrected by comparing to the real-time DMPS unit mass multiplied by concurrent calculated density from AMS species.*

*The mass ratio of AMS/(DMPS\*density) is 0.4 ~1.0 in all investigated VOC systems.''*

**Minor comments:**

**Q1.** P4/L88: suggest to change "even RH up to 90%" to "even with RH of up to 90%".

*Thanks for your kindness. This has been corrected in the update version.*

**Q2.** P8/L174 and L177: "don't" to "do not".

*Thanks for your kindness. This has been corrected in the update version.*

**Q3.** P8/L181: "100 Pa s in volatility"? Should it be viscosity?

*Yes, correct. This has been revised in the manuscript.*

**Q4.** P9/L184: add "and" before "SOA". Also in other places including L204, L234, L235, and L244

*Thanks for your kindness. These have been corrected in the update version.*

**Q5.** P11/L255: "are" to "is", and use ">" to denote the decreasing order?

*Thanks for your kindness. These have been corrected in the update version.*

**Q6.** P12/L266: "3.5h" to "3.5 h", and a few other places in the same paragraph.

*Thanks for your kindness. These have been corrected throughout the manuscript.*

**Q7.** P12/L270: add "was" before ">0.05".

*Thanks for your kindness. This has been corrected in the update version.*

**Q8.** P14/L319: "BVOC" to "BVOC-".

*Thanks for your kindness. This has been corrected in the update version.*

**Q9.** P14/L341: "need" to "needs".

*Thanks for your kindness. This has been corrected in the update version.*

**Q10.** Figures 1-3: I assume "[-]" in the axis titles means dimensionless. How about just state "dimensionless"?

*Thanks for your kindness. [-] have been removed in all figures.*

---

## Author Comment (AC2)

**Response to comments of anonymous referees # 2**

**General comments**

The manuscript is also fairly well written. I have a few comments for the authors to clarify, and recommend a Minor Revision before publication. The manuscript by Wang et al. reports particle rebound fractions (as a measure of particle phase states) measured for SOA particles produced from the photooxidation of single and binary VOC precursors in an environmental chamber. Compared to previous studies mainly focusing on organic aerosols, this study also examined the role of ammonium sulfate seed particles by varying the organic-to-inorganic mass ratio. The authors found that particle phase states to the first order depend on the RH and organic-inorganic ratio, while the VOC precursor type plays a relatively minor role. The experiments were carefully designed and executed. The paper was well written. However, I do have a few comments that need to be addressed before I can recommend publication.

*Many thanks to the reviewer for the comments and suggestions. We have improved the manuscript accordingly. Please find a point-by-point response below.*

**Specific comments:**

**Q1.** The authors claim o-cresol as a representative anthropogenic SOA precursor. The manuscript explains that the o-cresol was chosen mainly because of its OH reactivity similar to the biogenic precursors and its modest SOA yield. However, a major source of o-cresol can be biomass burning, either from manmade or natural sources. This is not discussed in the manuscript and I think it can be misleading to simply claim that o-cresol is anthropogenic.

*Good point. The description of the o-cresol has been rephrased as follows:*

*"Building on the McFiggans et al. (2019), we added a representative aromatic VOC (o-cresol) into initially designed binary mixtures of biogenic VOCs (isoprene & α-pinene). o-cresol can be emitted into atmosphere both directly from biomass burning (Coggon et al., 2019;Koss et al., 2018) and indirectly from the oxidation of toluene (e.g. motor vehicles, solvent use (Fishbein, 1985)). Therefore, anthropogenic source is one of the main contributors to the o-cresol but worth noting that the natural biomass burning can also be an important contributor."*

**Q2.** The presence of inorganic species can significantly alter the rebound curve, and the composition and O:C of organic species play a relatively minor role. I wonder if it is possible to develop a simple mixing role to predict the liquid-to-nonliquid phase transition? For example, is the transition RH related to liquid water content or hygroscopic growth factor?

*Thanks for your suggestion. During the experiments, we conducted concurrent hygroscopic measurement at 90% RH using a custom-made Hygroscopicity Tandem Differential Mobility Analyzer (HTDMA) for the same particle size with BF measurement. By using κ-Köhler theory, we calculated the growth factor (GF) at the RH of the BF, and linked the BF with the GF as shown in Figure 5 below. A brief description of HTDMA was added to the method, a new section 3.3 to the results to demonstrate the BF-GF relation, and the abstract and conclusions are revised to*

*include this result accordingly.*

[Figure]

*Figure 5. The relation of BF and hygroscopic GF of aerosol particles during six-hour photochemistry experiments in various VOC systems.*

*HTDMA description:*

*"Hygroscopic growth factor (GF) at 90% RH of submicron aerosol particles (75 ~ 250 nm) was recorded by a custom-made Hygroscopicity Tandem Differential Mobility Analyser (HTDMA) (Good et al., 2010), which can be used to calculate the GF at given RH using κ-Köhler approximation (Petters and Kreidenweis, 2007)."*

*Sec. 3.3 Mixing role of chemical composition and RH (GF) on BF*

*"As we know that the chemical composition and RH are key factors influencing aerosol water uptake at given size (Kreidenweis and Asa-Awuku, 2014). To discuss the mixing role of chemical composition and RH on BF, the GF at the RH of BF measurement was calculated from HTDMA measurement using κ-Köhler equation, and the relation between the BF and the GF was plotted in Figure 5. It can be seen that the BF showed a monotonic decrease from ~ 1 to ~ 0 with the increasing GF from ~ 1 to ~ 1.15 in all VOC systems. When the GF is larger than 1.15, the aerosol particles kept in liquid*

*phase (BF ~ 0). This evidence indicated the key role of aerosol water on the phase transition from the non-liquid to the liquid. The BF-GF relation of the varying multi-component aerosol particles including SOA and inorganic compounds is comparable with the previous study (Bateman et al., 2015). They measured the BF of the SOA from α-pinene, toluene and isoprene and found that the BF is almost 0 when the GF is larger than 1.15 (Bateman et al., 2015)."*

*One sentence has been added in the abstract to clarify this point:*

*"BF decreased monotonically with increasing hygroscopic growth factor (GF) and the BF was ~ 0 when GF was larger than 1.15."*

*Added sentences in the conclusion to clarify this point:*

*"Additionally, by combining the chemical composition and RH, we calculated the hygroscopic growth factor (GF) and found its key role in aerosol phase behaviour. The multicomponent aerosol particles were liquid in all VOC systems when the GF is higher than 1.15 at room temperature and transmitted from liquid to non-liquid when the GF decreased to ~ 1."*

**Q3.** Line 104-105: some studies do find that alpha-pinene SOA coating can influence the deliquescence of ammonium sulfate, although the effect is relatively smaller than isoprene SOA:

https://www.tandfonline.com/doi/full/10.1080/02786826.2010.532178

https://acp.copernicus.org/articles/12/9613/2012/acp-12-9613-2012.pdf

*Thanks for your helpful comment. The discussion of α-pinene SOA condensation on deliquescence of ammonium sulfate has been added:*

*"These results are partly consistent with similar studies using a different instrument (Hygroscopicity Tandem Differential Mobility Analyzer) (Smith et al., 2011;Smith et al., 2012). They found that the α-pinene SOA on ammonium sulphate seed can slightly shift the deliquescence and efflorescence RH by a few percent (Smith et al., 2011) while the isoprene SOA can significantly decrease the deliquescence and efflorescence RH depending on the organic fraction (Smith et al., 2012)."*

**Q4.** Were the rebound measurements performed for monodisperse or polydisperse aerosol particles? If polydisperse, were there particles smaller than the cutoff diameter of the impactor? This information might be provided in the literature cited in this paper, but it would be nice to briefly describe it here as well.

*Thanks for your suggestion. In this study, a DMA was used to select a monodisperse aerosol particles (100 ~ 200 nm) to measure the rebound fraction. The measured 50% transmission diameter ($D_a$) of the impactor is 84.9 ± 5.4 nm, which means that smaller aerosol particles (< $D_a$) will follow the gas stream and larger particles can hit to the impactor to effectively represent the bounce behaviour (Bateman et al., 2014). In this study, our selected sizes are larger than the $D_a$, so that we can derive reliable bounce behaviour. The clarification of the cutoff diameter of the impactor has been added to the method of the manuscript as follows:*

*"During the experiments, a Differential Mobility Analyzer (DMA) was used to select a monodisperse aerosol particles from chamber, with mobility diameter of 100 ~ 200 nm following the growth of the aerosol particles. The selected particle sizes are larger than the 50% transmission diameter of the used impactor (84.9 ± 5.4 nm) (Bateman et al., 2014) to ensure a reliable bounce fraction measurement."*

**References:**

Bateman, A. P., Belassein, H., and Martin, S. T.: Impactor Apparatus for the Study of Particle Rebound: Relative Humidity and Capillary Forces, Aerosol Science and Technology, 48, 42-52, 10.1080/02786826.2013.853866, 2014.

Bateman, A. P., Bertram, A. K., and Martin, S. T.: Hygroscopic Influence on the Semisolid-to-Liquid Transition of Secondary Organic Materials, The Journal of Physical Chemistry A, 119, 4386-4395, 10.1021/jp508521c, 2015.

Coggon, M. M., Lim, C. Y., Koss, A. R., Sekimoto, K., Yuan, B., Gilman, J. B., Hagan, D. H., Selimovic, V., Zarzana, K. J., Brown, S. S., Roberts, J. M., Müller, M., Yokelson, R., Wisthaler, A., Krechmer, J. E., Jimenez, J. L., Cappa, C., Kroll, J. H., de Gouw, J., and Warneke, C.: OH chemistry of non-methane organic gases (NMOGs) emitted from laboratory and ambient biomass burning smoke: evaluating the influence of furans and oxygenated aromatics on ozone and secondary NMOG formation, Atmos. Chem. Phys., 19, 14875-14899, 10.5194/acp-19-14875-2019, 2019.

Fishbein, L.: An overview of environmental and toxicological aspects of aromatic hydrocarbons II. Toluene, Science of The Total Environment, 42, 267-288, https://doi.org/10.1016/0048-9697(85)90062-2, 1985.

Good, N., Coe, H., and McFiggans, G.: Instrumentational operation and analytical methodology for the reconciliation of aerosol water uptake under sub- and supersaturated conditions, Atmos. Meas. Tech., 3, 1241-1254, 10.5194/amt-3-1241-2010, 2010.

Koss, A. R., Sekimoto, K., Gilman, J. B., Selimovic, V., Coggon, M. M., Zarzana, K. J., Yuan, B., Lerner, B. M., Brown, S. S., Jimenez, J. L., Krechmer, J., Roberts, J. M., Warneke, C., Yokelson, R. J., and de Gouw, J.: Non-methane organic gas emissions from biomass burning: identification, quantification, and emission factors from PTR-ToF during the FIREX 2016 laboratory experiment, Atmos. Chem. Phys., 18, 3299-3319, 10.5194/acp-18-3299-2018, 2018.

Kreidenweis, S., and Asa-Awuku, A.: Aerosol Hygroscopicity: Particle Water Content and Its Role in Atmospheric Processes, 331-361 pp., 2014.

McFiggans, G., Mentel, T. F., Wildt, J., Pullinen, I., Kang, S., Kleist, E., Schmitt, S., Springer, M., Tillmann, R., Wu, C., Zhao, D., Hallquist, M., Faxon, C., Le Breton, M., Hallquist, Å. M., Simpson, D., Bergström, R., Jenkin, M. E., Ehn, M., Thornton, J. A., Alfarra, M. R., Bannan, T. J., Percival, C. J., Priestley, M., Topping, D., and Kiendler-Scharr, A.: Secondary organic aerosol reduced by mixture of atmospheric vapours, Nature, 565, 587-593, 10.1038/s41586-018-0871-y, 2019.

Petters, M. D., and Kreidenweis, S. M.: A single parameter representation of hygroscopic growth and cloud condensation nucleus activity, Atmos. Chem. Phys., 7, 1961-1971, 10.5194/acp-7-1961-2007, 2007.

Smith, M. L., Kuwata, M., and Martin, S. T.: Secondary Organic Material Produced by the Dark Ozonolysis of α-Pinene Minimally Affects the Deliquescence and Efflorescence of Ammonium Sulfate, Aerosol Science and Technology, 45, 244-261, 10.1080/02786826.2010.532178, 2011.

Smith, M. L., Bertram, A. K., and Martin, S. T.: Deliquescence, efflorescence, and phase miscibility of mixed particles of ammonium sulfate and isoprene-derived secondary organic material, Atmos. Chem. Phys., 12, 9613-9628, 10.5194/acp-12-9613-2012, 2012.